# Assessment of Phytotoxicity in Untreated and Electrochemically Treated Leachates through the Analysis of Early Seed Growth and Inductively Coupled Plasma-Optical Emission Spectroscopy Characterization

Alfredo Martínez-Cruz and María Neftalí Rojas-Valencia *

Institute of Engineering, National Autonomous University of México, External Circuit, University City, Coyoacan Delegation, Mexico City 04510, Mexico; amartinezcr@iingen.unam.mx
* Correspondence: mrojasv@iingen.unam.mx; Tel.: +52-(55)-5633600 (ext. 8663)

**Abstract:** The treatment of stabilized leachates with high refractory organic matter content, which are over 10 years old, presents a challenge. This study explored the potential of electro-coagulation (EC) and electro-oxidation (EO) treatment systems to address this issue. The objective of this study was to investigate the phytotoxicity of the proposed treatment system on seed growth and examine possible relationships between phytotoxicity results and the characterization of leachates, effluents, soil, and radicles. Phytotoxicity tests were conducted on seeds of *Lactuca sativa*, *Cucumis sativus*, and *Phaseolus vulgaris*, using Inductively Coupled Plasma-Optical analysis. The evolution of organic matter was monitored by fractionating the chemical oxygen demand (COD) and humic substances. The biodegradability index increased from 0.094 in raw leachate to 0.26 and 0.48 with EC and EO, respectively. Removal rates of 82%, 86%, 99%, and 81% were achieved for COD, dissolved organic carbon, color, and ammoniacal nitrogen, respectively. The biodegradable COD increased from 26% in raw leachate to 39% in the EC process and 58% in the EO process effluent. The proposed treatment system successfully broke the aromatic structures of the humic substances present in the raw leachate, thereby increasing the content of biodegradable material. Phytotoxicity tests revealed that the proposed treatment system significantly reduced the phytotoxicity of the generated effluents.

**Keywords:** circular economy; electro-coagulation; electro-oxidation; mature leachates; organic matter; phytotoxicity





## 1. Introduction

The overall generation of municipal solid waste (MSW) has steadily increased because of urbanization, increased commerce, population growth, and economic development [1]. By the end of the next three decades, it is projected that global MSW generation will reach 3.4 billion tons, up from 2.01 billion tons generated in 2016 [2]. The prevalent approach for MSW disposal in developing and underdeveloped countries is landfilling [3]. Given the moisture content and seasonal fluctuations in MSW, the production of hazardous liquid, known as landfill leachate [1], is a natural by-product of landfills (hereinafter referred to as "leachates" in this document).

From the viewpoint of environmental pollution, the crucial components of leachates include organic pollutants, such as volatile fatty acids (acetic, propionic, butyric, caproic, valeric, and heptanoic acid), humic substances (HS), and inorganic pollutants, including heavy metals, ammoniacal nitrogen (NH₃-N), sulfates, sodium, magnesium, calcium, and other xenobiotic compounds, such as aromatic hydrocarbons and halogenated organic compounds [4].

Untreated leachates have the potential to inflict severe and irreversible harm on both aquatic and terrestrial ecosystems [1]. The categorization of leachates is based on their age,

which can be classified as young (less than 5 years), intermediate (5–10 years), or stabilized (>10 years), as reported by Kow et al. [5]. For the biological treatment of leachates, the ideal $BOD_5/COD$ ratio is >0.5 [6], but previous investigations conducted by our group have determined that the biodegradability index ($BOD_5/COD$ ratio) of the leachates used in this project (Bordo Poniente landfill, Mexico City, México) was $0.094 \pm 0.03$ [7,8].

Leachates are among the most complex forms of wastewater, and their treatments can be classified into three categories: biological (aerobic/anaerobic), physical and chemical (sedimentation/flotation, coagulation/flocculation, adsorption, and chemical oxidation), and combined or hybrid treatment methods [9,10]. It is worth noting that, as the age of the leachate increased, the concentration of recalcitrant organic matter increased significantly. Conventional treatment methods have proven ineffective for treating recalcitrant organic matter, leading to the growing popularity of advanced oxidation processes (AOPs) for leachate treatment [10]. It is of utmost importance to thoroughly understand the characteristics of leachates to evaluate their potential to cause harm to the environment and to determine the most effective treatment methods. Although these physicochemical treatment methods are effective, they are not cost-efficient and produce a significant amount of sludge that must be processed further, as reported by Ghosh et al. [4] and Kamaruddin et al. [3].

In contrast, electrochemical technologies have the potential to become dominant solutions for the removal of biorefractory substances [11,12]. Among electrochemical technologies, electro-oxidation (EO) has been demonstrated to be effective in significantly reducing recalcitrant organic compounds [13–15]. The treatment of stabilized leachates via EO can be a challenging endeavor because of the presence of colloidal and hydrophobic humic substances, which can hinder the efficiency of the process and increase the energy costs. However, by implementing an electro-coagulation (EC) process as a preliminary treatment, it is possible to separate these humic substances. This results in a more efficient treatment process and elimination of interference from these colloidal species [7,16–18]. The EC process has been demonstrated to be effective in treating various complex wastewaters, including leachate, as reported in several studies [7,19].

Assessment of the toxic effects of leachates can be conducted through various toxicity tests that utilize diverse bioassays, such as microorganisms (Ames mutagenicity test, luminescent bacteria test, etc.), plants (seed germination study, micronucleus test, comet assay, etc.), and animals (e.g., mortality and contact tests). These tests were performed using standardized protocols to evaluate the potential toxicity of leachates to living organisms [1,20]. Plant bioassays, such as seed germination studies, are commonly employed to evaluate the toxicity of hazardous waste, including leachates. These tests are particularly sensitive to the toxic effects of substances such as low-molecular-weight acids (acetic, propionic, butyric, caproic, valeric, and heptanoic acid), $NH_3$-N, heavy metals, and salts present in leachates. To quantify the toxicity of leachates, both the germination index (GI) and root inhibition of the seeds can be measured. Furthermore, it should be noted that seed germination studies are not only rapid but also economical, compared to other toxicity assessment methods [1].

Although numerous studies have explored leachate treatment [8,20–36], most studies have focused on changes in parameters, such as chemical oxygen demand (COD) and biochemical oxygen demand ($BOD_5$), without considering the impact of treatment on the toxicity levels of the resulting effluents. Based on the limited data available, it is difficult to establish a correlation between leachate characteristics, effluent leachate results subsequent to the chosen treatment, and potential modifications in toxicity levels in both leachates and effluents. Despite the dearth of reports on the toxicity of leachates to seeds using germination and root inhibition assays, the root growth assay continues to be a highly efficacious method for toxicity evaluation.

The assessment and quantification of toxic chemical elements, particularly those of utmost concern, are crucial components of chemical analysis. Inductively coupled plasma-optical emission spectroscopy (*ICP-OES*) has emerged as the preferred method

for elemental analysis, owing to its ability to simultaneously measure multiple elements. This method employs inductively coupled plasma to produce excited atoms and ions that emit the characteristic electromagnetic radiation [37]. *ICP-OES* is known for its exceptional accuracy, precision, and low detection rate [38]. There has been limited prior research establishing a correlation between phytotoxicity and comprehensive analysis by *ICP-OES* in both the influents and effluents of a treatment system, such as that proposed in this project.

According to Derdera and Ogato [39], the global cost of MSW management is projected to reach USD375.5 billion by 2025. This issue is particularly pressing in developing countries where sustainable waste management is a major concern [40]. Urban solid waste management is a complex problem encompassing environmental, economic, institutional, social, and political aspects [41]. To address this challenge, efforts are needed to increase funding, raise public awareness, invest in infrastructure, and build expertise [39]. This project is particularly relevant in the context of expertise creation and highlights the importance of this initiative. The proposal of treatment systems that include organic matter monitoring is essential for understanding the processes involved and making recommendations for improvement. This specialized knowledge, combined with effective MSW management, represents a significant effort to preserve the environment.

The circular economy paradigm aims to transform various sectors to adopt its principles as an alternative to the traditional linear economy model. Natural resources are utilized to produce goods, resulting in waste that must be disposed of [42]. In contrast, a circular economy emphasizes the recovery and reuse of resources [43]. There is growing interest in promoting a more circular economy, with the ultimate goal of minimizing waste [44].

The scarcity of resources is driving a shift towards more sustainable production systems, including the recovery of resources from MSW. Although landfills are the most common method of waste management in developing countries, leachate treatment systems must be integrated into this circular economy approach to ensure resource recovery and sustainability [45]. The proposed EC-EO treatment system for leachates seeks to reduce resource usage compared with conventional advanced oxidation treatments, which rely on the use of chemicals. Additionally, evaluating the phytotoxicity of treated effluents can pave the way for future research into the potential reuse of treated leachates, thereby promoting a circular economic approach to MSW management.

This study aimed to investigate the phytotoxicity of early seed growth in raw leachates and raw leachates treated with an EO-EC system. Previously, the optimal conditions for the EC-EO treatment system were identified to remove organic matter, as measured by COD. Additionally, an investigation was conducted on the raw leachates and generated effluents, which included an examination of any potential connections between the phytotoxicity results and the *ICP-OES* characterization.

## 2. Materials and Methods

The methodology employed in this study encompasses well-defined stages. Raw leachates were collected and analyzed. Subsequently, the leachates were treated using the proposed EC and EO systems. Operational parameters were optimized for each process to attain maximum elimination of organic matter, as measured by chemical oxygen demand (COD). Under these optimal conditions, organic matter content was assessed through humic substance fractionation and COD analysis, and phytotoxicity tests were conducted to evaluate the early growth of the selected seeds in the resulting effluents (*Lactuca sativa*, *Cucumis sativus* and *Phaseolus vulgaris*). Finally, in addition to the phytotoxicity tests, inductively coupled plasma-optical emission spectroscopy (*ICP-OES*) analysis was performed on the garden soil and the roots generated in the phytotoxicity tests. A visual representation of this methodology is shown in Figure 1.

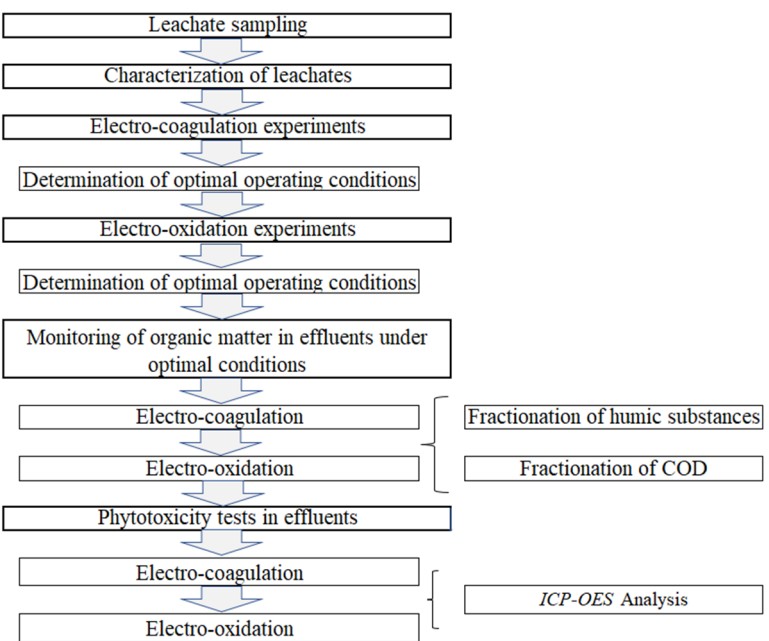

**Figure 1.** Diagram of the methodology developed in the project.

### 2.1. Landfill Leachate Collection and Characterization of Leachates and Effluents

Leachate samples were collected from the Bordo Poniente Landfill Stage III. This landfill is located southwest of the former Lake Texcoco, five kilometers from Mexico City's international airport (longitude 99°00 14.51 and 99°02 36.21 west; latitude 19°26 09.36 and 19°29 09.22 north), has an area of 670 ha, and was operated from 1985 to 2011 (Figure 2). This landfill is the largest in Latin America and contains 76 million tons of urban solid waste [7]. Leachate samples (80 L) were collected and transferred to plastic containers, which were subsequently stored in a refrigerator at 4 °C until analysis.

American Standard Test Methods were used to determine the dissolved organic carbon (DOC) and color [46,47]. Standardized American Public Health Association APHA methods were used to determine pH, electrical conductivity, COD, $BOD_5$, chlorides, and $NH_3$-N [48]. Fractions of COD (via the determination of biodegradable COD by aeration and soluble COD by coagulation with zinc sulfate [49,50]) and humic substances (via the separation of humic acids by acidification and fulvic acids by adsorption) were determined [51].

### 2.2. Leachate Treatment System

In this study, a two-stage treatment system was developed and implemented. The initial stage involved the application of a process known as EC to the raw leachate. This was followed by the EO process, which was optimized using response surface methodology and a fractional orthogonal design to achieve the maximum removal of organic matter. The goal of this second stage was to produce the final effluent with the highest possible degree of organic matter removal. The details of this procedure have been documented in a previous study conducted by the same research team [7].

#### 2.2.1. Electro-Coagulation Process

The EC process utilized an experimental design, which consisted of a fractional orthogonal type L9 (34) with nine runs, three levels per factor, and three replicates [52,53]. Kolmogorov-Smirnov and Levene tests were conducted to assess the normality and equality of variances [52]. The dependent variable, measured as COD, was the removal of organic matter. The dependent variables and their ranges for this study were determined based on previous research in the field of leachates [54–57]. The specified ranges were as follows: the current density ranged from 13.8 mA cm$^{-2}$ to 30.5 mA cm$^{-2}$, the stirring rate ranged from 0 to 200 rpm, and the pH ranged from six to eight. With regard to the duration of

the process, considering the recommendations in the literature [58,59], it was decided to establish times of 120 min. To streamline the experimental procedure, it was suggested that the EC process maintain a fixed distance of 1 cm between the electrodes. The dependent variables were pH, current density, and agitation.

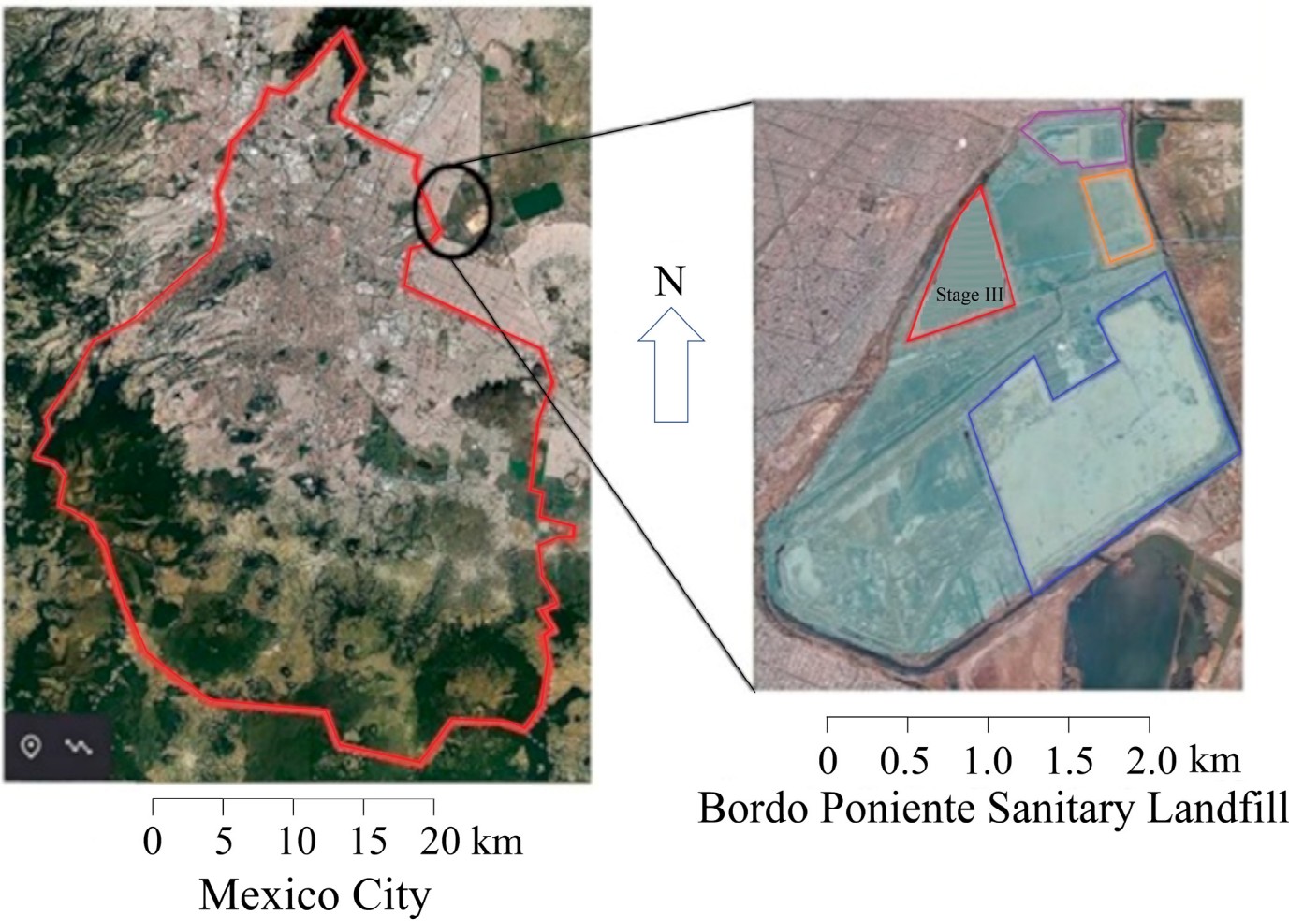

**Figure 2.** Study area. Bordo Poniente landfill (Google Maps).

The reactor used for the EC process had dimensions of 15 cm × 10 cm × 15 cm, with a volume of 1.5 L, and comprised an Fe anode and a stainless steel cathode, each measuring 10 cm × 10 cm × 0.4 cm, with an effective area of 160 cm$^2$ and a distance between electrodes of 1 cm. EC electrodes were connected in monopolar mode, in parallel to a digital DC power supply (Steren PRL258, Steren®, Mexico City, Mexico) in batch operation mode.

2.2.2. Electro-Oxidation Process

The experimental design for EO was a fractionated orthogonal approach with nine runs, three levels per factor, and three replicates. The dependent variable, measured as COD, was the removal of organic matter. A constant flow rate of 1.2 L min$^{-1}$ for the EO reactor was established. The dependent variables were pH and current density, the distance between the electrodes was adjusted, and NaCl was added as an additional electrolyte.

The ranges of the dependent variables were established through preliminary experiments and a literature review [49,60,61]. The specified ranges were as follows: the current density ranged from 16.6 mA cm$^{-2}$ to 50 mA cm$^{-2}$, the electrode spacing ranged from 0.5 to 1 cm, the concentration of the additional electrolyte (NaCl) ranged from 0 to 2 g L$^{-1}$, and the pH ranged from six to eight. The duration of the process, considering the recommendations in the literature [60,61], was established as 60 min.

The EO reactor had dimensions of 12 cm × 10 cm × 12 cm, with a volume of 1.5 L, and comprised a boron-doped diamond (BDD) anode and a stainless steel cathode, each measuring 7 cm × 5 cm × 0.1 cm, with an effective area of 60 cm². The EO process employed a filter press arrangement in an upper frame that held the electrodes, with the option of varying the distance between the electrodes at 0.5, 1.0, and 1.5 cm, and a continuous flow of input and output using a recirculation vessel at a rate of 1.5 L min⁻¹. EO electrodes were connected in monopolar mode, in parallel to a digital DC power supply (Steren PRL258, Steren®, Mexico City, Mexico) in batch operation mode. Figure 3 shows the design of the proposed treatment system.

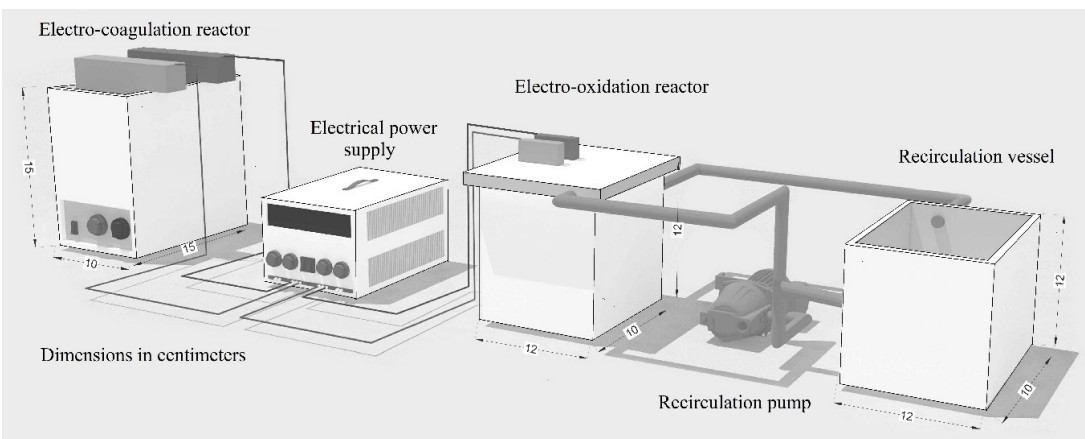

**Figure 3.** Experimental setup used in the experimentation of this project.

*2.3. Evaluation of Phytotoxicity*

*Lactuca sativa* seeds are widely recognized as standardized seeds for phytotoxicity testing and have been endorsed by the Organization for Economic Cooperation and Development [62], United States Environmental Protection Agency [63], and American Society for Testing and Materials [64]. The inclusion of *Cucumis sativus* and *Phaseolus vulgaris* in the phytotoxicity tests was intended to provide data on the potential effects of contaminants on plant communities in and around functioning and closed landfills. These seeds were selected because of their sensitivity to environmental stress, ease of handling, relatively short life cycle, and the ability to provide consistent and reproducible results. Additionally, they are commonly cultivated in areas near landfills in Mexico, making them particularly relevant for horticultural considerations.

The phytotoxicity testing method utilized the International Organization for Standardization ISO 11269-2, Organization for Economic Cooperation and Development OECD Test Guideline 208, and the United States Environmental Protection Agency Test Guidelines OPPTS 850.4200 [62,63,65] to evaluate the effects of different concentrations of leachates and effluents on the growth of lettuce (*Lactuca sativa*), cucumber (*Cucumis sativus*), and bean (*Phaseolus vulgaris*) in a garden soil substrate. Gardening land was purchased from a local specialty store in Mexico City, Mexico. The seeds were purchased from 8 g packs in Rancho Los Molinos ® (Mexico City, Mexico).

Upon identifying the optimal operating conditions for the maximum removal of organic matter, as measured by COD, in both EC and EO processes, three types of leachates were generated. These include crude leachate, EC effluent, and EO effluent. A mixture of these three leachates and tap water was used to prepare the dilutions employed in the experimental protocol. The seeds were subsequently placed within the treated soil and assessed for growth and development seven days after the emergence of 50% of the seedlings in the control group. Measurements and observations were compared to those of the untreated control.

Following the test, all plants, including their roots, stems, and leaves, were carefully removed from the pots. The thin and thick roots were thoroughly washed with tap water

to remove any soil residue. All the samples were subsequently dried in an oven at 75 °C, after which the dry matter content was determined.

The seeds were subjected to a formal tone and soaked in tap water for two hours prior to the experiment. The seeds were then evenly distributed in germination containers, which consisted of ten seeds of *L. sativa* and five seeds each of *C. sativus* and *P. vulgaris*, in a container filled with commercial garden soil that had been soaked with 5 mL of the evaluated effluent. The germination vessels, made of polypropylene, had 12 cells with dimensions of 0.3 × 0.3 m, and the research was conducted with 12 replicates (Figure 4). The germination vessels were kept in a greenhouse under a translucent plastic film cover that allowed the passage of light but not rain.

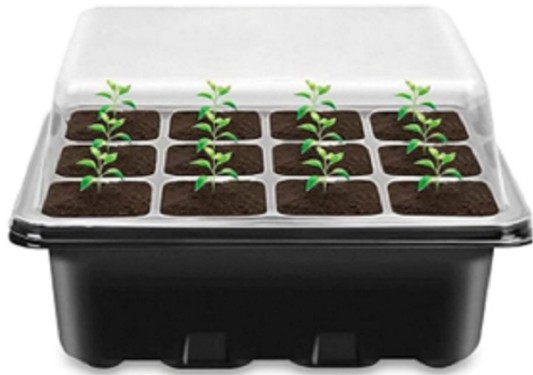

**Figure 4.** Germination tray used in the phytotoxicity experiments.

The germination rates (Equation (1)), radicle length (Equation (2)), growth inhibition of biomass (Equation (3)), and the germination index (Equation (4)) were subsequently determined as part of the phytotoxicity test.

$$\% \text{ Germination rate} = ((\text{Number of seeds germinates})/(\text{Total of seeds})) \times 100 \quad (1)$$

$$\% \text{ Growth inhibition} = ((\text{Radicle length control} - \text{Radicle length dilution})/(\text{Radicle length control})) \times 100 \quad (2)$$

$$\% \text{ Growth inhibition} = ((\text{Average weight in control} - \text{Average weight in treatment})/(\text{Average weight in control})) \times 100 \quad (3)$$

$$\text{Germination index} = (\text{Radicle length dilution}/\text{Radicle length control}) \times (\text{Germinated seed dilution}/\text{Germinated seed control}) \quad (4)$$

### 2.3.1. Experiment Design

The experimental design was a factorial design with 12 replicates. The independent variables were the seed type (*L. sativa*, *C. sativus*, and *P. vulgaris*), the dilution of the leachate and treated leachates (0, 2.5, 10, 30, 60, and 90%), the pH (unadjusted pH and a neutral pH of 7.0 using citric acid or NaOH 0.1 N), and the leachate (row leachate, EC effluent, and EO effluent). To avoid the effects of oxidation on the organic matter in the samples, citric acid (20 wt.) was used, following the recommendations of agroecological companies [66].

The response variables were the germination rate, growth inhibition (radicle length), growth inhibition (biomass), and germination index. The half maximal effective concentration (EC50) was calculated from the dose-response relationship between the germination rate and the dilution of the leachates or effluent, using probit analysis with confidence limits of 95%.

MINITAB Release 18 statistical software was used to conduct a thorough analysis supported by statistical methods. The results are presented as the mean ± standard deviation, and the statistically significant difference between the control and treated groups was determined using one-way and two-way analyses of variance (ANOVAs).

### 2.3.2. Transfer Coefficient and Enrichment Coefficient

In addition to the phytotoxicity tests, the transfer coefficient (TC) and enrichment coefficients (EnC) were determined. The TC and EnC were calculated to evaluate the capacity of the seed species to concentrate soil metals. The TC was determined as the ratio of the metal concentration in the root to its content in the soil (mg kg$^{-1}$) (Equation (5)) [67].

$$\text{TC = Plant root metal concentration/Soil metal concentration} \tag{5}$$

The EnC quantifies the ability of the plant to accumulate soil metals. The EnC is a metric used to evaluate the degree of soil contamination and metal accumulation in plants cultivated in polluted soil, relative to plants and soil in a control environment. This was determined by comparing the metal concentrations in the contaminated soil to those in the uncontaminated soil (Equation (6)) [68].

$$\text{EnC = Plant root metal concentration using leachate/Plant}$$
$$\text{root metal concentration in control sample} \tag{6}$$

### 2.4. Analysis Using Inductively Coupled Plasma-Optical Emission Spectroscopy (ICP-OES)

Undeniably, the significance of resource and time optimization is paramount. Consequently, the roots obtained from the effluent of the complete treatment system (EC-EO) underwent ICP-OES analysis. Additionally, the analysis was extended to include the garden land used in the tests. The methodology is described as follows:

### 2.4.1. Chemical Reagents and Materials and Preparation of Samples

High-purity (99.99%) argon was used as the plasma, auxiliary, and nebulizer gas. Nitric acid and hydrogen peroxide, both analytical research grade, were obtained from Merck ® (Darmstadt, Germany) for use in this experiment. All glassware used in the experiment was of "A" grade quality and was calibrated. A micropipette with a calibrated range of 100-1000 μL was used in the experiment.

The quantification of roots and soil was conducted, with approximately 0.4 g of each being measured. Subsequently, 5 mL of leachate samples were collected. Nitric acid and hydrogen peroxide were added to the digestion tube at a volume ratio of 5 mL. An assisted microwave digestion technique, as outlined by Stadler and Michaelis [69] was used to heat the organic matter in the samples to 120 °C for 30 min. After digestion, the samples were filtered through Whatman No. 41 paper filter in volumetric flasks with a capacity of 25 mL. The volume was adjusted using 5% weight nitric acid solution to achieve the desired quantity.

### 2.4.2. Inductively Coupled Plasma-Optical Emission Spectroscopy (*ICP-OES*) Equipment

The equipment used was the Agilent Technologies 5100 ICP-OES (Santa Clara, CA, USA), with a concentric nebulizer, a gas flow rate of 12 L/min, and an auxiliary gas flow rate of 0.7 L min$^{-1}$ (argon). The radiofrequency power used was 1.2 kW, and the pump speed was 12 rpm. A charge-coupled device (CCD) detector and 21 CFR 11 version 4.1.0 software were used for data acquisition and processing.

For the calibration curve preparation of the instrument, a standard multi-element solution of 1000 μg mL$^{-1}$ K and 100 μg mL$^{-1}$ for the remaining elements were used (Quality Control Standard 26, hps ®, Charleston, SC, USA). During the ICP-OES studies, the vertical height of the plasma was maintained at a constant level of 7 mm. The sample uptake time was set at 30.0 s, followed by a 5-s delay and a 10-s rinse time. The initial stabilization period was 10 s, and the time between replicate analyses was fixed at 5 s. These parameters were maintained throughout the study period.

## 3. Results and Discussion

### 3.1. Treatment System under Conditions of Increased Removal of Organic Matter and Characterization of Leachates and Effluents

The optimal conditions for achieving the significant removal of organic matter, as measured by COD, during EC were a current density of 23.3 mA cm$^{-2}$, stirring at 100 rpm, and a pH of 7. These conditions resulted in COD, DOC, color, and NH$_3$-N removal rates of 63, 69, 94, and 50%, respectively. The most effective conditions for EO were an NaCl concentration of 1 g L$^{-1}$, electrode distance of 0.75 cm, current density of 33.3 mAcm$^{-2}$, and pH of 7. Under these conditions, COD, DOC, color, and NH$_3$-N were removed at rates of 82, 86, 99, and 81%, respectively (compared to raw leachate).

In the EC, the results of the Kolmogorov-Smirnov test indicated that the data were normally distributed ($p = 0.42 > 0.05$), as did the Levene test for the equality of variances for the experimental data ($p = 0.58 > 0.05$). For EO, the results of the Kolmogorov-Smirnov test and Levene test indicated that the experimental data did meet the assumptions of normal distribution and equal variances, respectively. Specifically, the $p$-value for the Kolmogorov-Smirnov test was 0.42, which is greater than the significance level of 0.05, suggesting that the data may be normally distributed. Similarly, the $p$-value for the Levene test was 0.64, which was also greater than the significance level of 0.05, indicating that the variances of the experimental data may be equal.

The attributes of the raw leachate, EC, and EO effluents are listed in Table 1. The pH of the raw leachates was found to be 8.4 ± 0.1, which aligns with the results reported in other studies on mature leachates. Poblete and Pérez [70] reported a pH value of 8.9 in mature leachates. Typically, mature leachates have a pH greater than seven because of the microbiological reduction of volatile fatty acids (acetic, propionic, butyric, caproic, valeric, and heptanoic acid) and a decrease in the leaching force during the methanogenic phase [71,72]. These high pH values are consistent with the stages that MSW undergoes in landfills, including acetogenic and methanogenic stages [73]. During EC, the pH of the effluent increases because of the presence of hydroxyl groups [57]. The decrease in pH during EO was attributed to the generation of H$^+$ ions [55].

**Table 1.** Characterization of raw leachates and effluents generated.

| Parameter | Raw Leachate | Electro-Coagulation Effluent | Electro-Oxidation Effluent |
|---|---|---|---|
| pH | 8.4 ± 0.1 | 9.5 ± 0.1 | 5.6 ± 0.1 |
| Biodegradability index | 0.094 | 0.26 | 0.48 |
| Electrical conductivity | 8.5 ± 1 | 2.2 ± 0.5 | 1.7 ± 0.4 |
| Chlorides | 6.7 ± 0.1 | 3.6 ± 0.1 | 1.9 ± 0.4 |
| BOD$_5$ | 0.32 ± 0.01 | 0.338 ± 0.01 | 0.288 ± 0.01 |
| Color | 3200 ± 90 | 200 ± 10 | 20 ± 1 |
| DOC | 1.2 ± 0.2 | 0.36 ± 0.01 | 0.16 ± 0.01 |
| NH$_3$-N | 0.66 ± 0.03 | 0.33 ± 0.01 | 0.12 ± 0.01 |
| Total COD | 3.4 ± 0.1 | 1.3 ± 0.5 | 0.6 ± 0.01 |
| Soluble COD | 1.77 ± 0.1 | 0.97 ± 0.2 | 0.4 ± 0.1 |
| Biodegradable COD | 0.87 ± 0.04 | 0.49 ± 0.1 | 0.35 ± 0.01 |
| Non-biodegradable soluble COD | 0.89 ± 0.04 | 0.48 ± 0.02 | 0.052 ± 0.02 |
| Humic acid | 1.94 ± 0.04 | 0.61 ± 0.03 | 0.37 ± 0.02 |
| Fulvic acid | 0.77 ± 0.03 | 0.29 ± 0.01 | 0.13 ± 0.01 |
| Hydrophilic fraction | 0.87 ± 0.04 | 0.34 ± 0.01 | 0.1 ± 0.01 |

Units in g L$^{-1}$, except for the color (Pt-Co U) and electrical conductivity (mS cm$^{-1}$). The pH and biodegradability indices are dimensionless. Results are expressed as mean ± standard deviation.

The biodegradability index value obtained for the raw leachates was 0.094, which is less than 0.1. This indicates that the leachate sampled at the Bordo Poniente landfill was of the mature type, as confirmed by Naveen et al. [74]. The results obtained were consistent with the closure of Stage III of the landfill, which has been in place since 1994 and has an

age well above 10 years. According to Abunama et al. [75], a biodegradability index of less than 0.1 signifies a substantial quantity of non-biodegradable organic matter. This finding is consistent with the non-biodegradable COD level of 74% identified in the crude leachate in this study. The treatment system resulted in changes in the biodegradability index. For EC, a value of 0.26 was obtained, and for the EO process, it reached 0.48. In the case of EC, the destabilization of colloidal material is achieved by coagulating organic matter (mainly humic substances). Regarding the EO process, the increase in the biodegradability index is justified by the removal of organic matter caused by the action of ●OH radicals formed at the BDD anode.

The raw leachate exhibited a color value of $3200 \pm 90$ U Pt-Co. The color of leachates is commonly associated with the presence of organic substances, particularly during the final phase of degradation, which is primarily composed of humic substances, such as humic and fulvic acids [76,77]. The discoloration observed in the EC process (95%) is primarily attributable to the removal of humic substances, which constitute the majority of the organic matter present in mature leachates. Therefore, the reduction in color is indicative of a concomitant decrease in the aforementioned humic substances caused by EC [51]. The EO process effectively reduced the remaining 90% of the color, resulting in a final reading of 23 Pt-Co units. This significant decrease can be attributed to the removal of humic substances, particularly fulvic acids, which are more challenging to remove through EC.

The concentration of $NH_3$-N in the raw leachates collected was 658 mg $L^{-1}$. This result can be ascribed to the biological degradation of amino acids and other nitrogenous organic substances present in MSW; a process previously documented by Abunama et al. [75]. EC removed 50% of $NH_3$-N. According to [78], the use of EC results in the removal of $NH_3$-N, which can be attributed to the increase in temperature and pH levels. These changes disrupted the equilibrium present in the $NH_3$-$NH_4^+$ leachates, promoting the release of $NH_3$-N as a gas into the atmosphere. According to Fudala-Ksiazek et al. [79], the decrease in $NH_3$-N during EO is attributed to its indirect oxidation resulting from the formation of reactive chloride oxidants, including $Cl_2$ and HOCl, which accounts for 81% of the reduction.

In the EC process, the destabilization of colloidal material caused by the presence of iron hydroxides is responsible for the removal of organic matter, as measured by the chemical demand for oxygen and dissolved organic carbon. Similarly, in the EO process, complete oxidation of a portion of the organic material to $CO_2$ is achieved through the action of ●OH radicals formed at the anode, resulting in the removal of organic matter [80].

In the raw leachates examined, the fractions of humic substances, as determined by COD, were as follows: humic acids in $1.94 \pm 0.04$ g $L^{-1}$, comprising 54% of the total, and fulvic acids in $0.77 \pm 0.03$ mg $L^{-1}$, comprising 22% of the total. Previous studies reported comparable results for mature leachates. Specifically, Dia et al. [51] reported 27% humic acid and 42% fulvic acid, collectively accounting for 76% of the total COD. The presence of humic substances in the leachates confirmed their maturity.

Until the 1990s, "humic substances" (including humic fractions) were considered to be unique, new, macromolecular compounds; it is now understood that they are in fact varying mixtures of identifiable molecules that can be attributed to plant and microbial sources [81,82]. Humic substances are composed of large molecules with a diverse array of functional groups such as benzene, phenols, phthalates, and polycyclic aromatic hydrocarbons (PAHs). These substances are found in carbohydrates, phenols, benzene, and lignin. Humic acid (HA) is primarily derived from monocyclic and polycyclic aromatic hydrocarbons, as well as heterocyclic hydrocarbons. The distinct differences between HA and fulvic acid (FA) suggest that the binding properties of these molecules can be influenced by their molecular structures, in addition to their carboxyl and phenolic groups [83]. It has been demonstrated that the binding properties of HA and FA are highly comparable. The potential for humic substances to form complexes with metals and their adsorption onto mineral surfaces may give rise to environmental issues. The rate at which such complexation occurs

may diminish when humic substances are reduced. Furthermore, the reduction in humic substances may facilitate the application of biological systems for treating effluents, as it suggests a decrease in the presence of recalcitrant substances.

In the EC process, it was observed that the biodegradable fraction of COD increased from 26% to 39%, owing to the elimination of hydrophobic substances from humic acids [51]. Conversely, the particulate fraction of COD decreased from 48 to 23%, which could be attributed to the removal of colloids [84]. The EO process resulted in a noticeable increase in biodegradable COD from 39% to 58%. This increase can be attributed to indirect EO mediated by ●OH radicals, which oxidize nonbiodegradable organic matter. Consequently, the soluble COD fraction decreased from 77% to 67%, primarily because of oxidation of the hydrophilic fraction during the process. Additionally, the biodegradable soluble COD fraction significantly increased from 51% to 87% because of the action of ●OH radicals that oxidize recalcitrant organic matter, such as humic and fulvic acids [60,85].

Mechanisms Involved in the Processes

In the process of EC, coagulant ions are generated in situ, which involves a multitude of synergistic mechanisms to eliminate pollutants from wastewater. EC consists of three stages: the dissolution of the sacrificial anode through the hydrolysis of metal ions to form hydroxide complexes, the destabilization of contaminants, and the adsorption of contaminants in precipitates to form flocs [86]. The following formal mechanism has been proposed for this process, as summarized in Equation (7) [87]. According to Yildiz et al. [87], $Fe(OH)_3$ particles remain suspended in aqueous solutions, where they can remove contaminants through agglomeration or electrostatic attraction, followed by coagulation.

$$4\ Fe_{(s)} + 10\ H_2O_{(l)} + O_{2(g)} \rightarrow 4\ Fe(OH)_{3(s)} + 4\ H_{2(g)} \tag{7}$$

In the EO process, two mechanisms are necessary to account for the removal of organic charges:

1.  In the initial stage of direct EO, organic pollutants diffuse from the electrolyte to the anode surface where they are adsorbed. Subsequently, the organic compounds were oxidized at the anode surface through electron transfer, as demonstrated in Equation (8), where "R" represents the organic pollutant and "P" represents the oxidized organic pollutant [85]. Direct EO leads to the formation of ●OH radicals adsorbed on the anode surface, which further oxidize the organic compounds through indirect electrolysis [85].

$$R \rightarrow P + e^- \tag{8}$$

2.  Indirect EO is a process that takes place at a potential higher than the "water stability" potential, resulting in the generation of hydroxyl radicals (●OH). These radicals adsorb onto the anode surface and prove to be efficient in the oxidation of organic compounds, including the degradation of recalcitrant aromatic substances such as humic substances present in stabilized leachates. A model has been proposed to elucidate the degradation of organic compounds using BDD as an anode, which is described by Equation (9) [88].

$$BDD(●OH) + R \rightarrow BDD + m\ CO_2 + n\ H_2O + H^+ + e^- \tag{9}$$

Although the study demonstrated that EC and EO processes were effective in removing organic matter (COD), the elimination of underestimated contaminants, including emerging pollutants, pharmaceuticals, personal care products, and commercial organic substances such as perfluoroalkyl acids, was not within the scope of this project [89]. Future research should consider the potential environmental problems caused by the migration of these pollutants and propose specific treatment systems to address them.

### 3.2. Evaluation of Phytotoxicity

### 3.2.1. Research on Seed Germination

By analyzing the data from the early germination of seeds, it was possible to gain insight into the phytotoxicity of leachates and effluents. If the phytotoxicity was high, seed germination was inhibited; however, if the phytotoxicity was low, seed germination occurred, and root growth could be measured. In this study, control experiments were conducted, and the seeds of *C. sativus* germinated on the second day, while the seeds of *L. sativa* and *P. vulgaris* germinated on the third day. The germination rates are shown in Table 2. At the control dilution, the germination rates of all three seeds were higher than 90% in all experiments. The average germination values were higher in leachates with neutral pH than those without adjustment ($p < 0.05$), and no statistically significant differences were observed between the three seeds evaluated ($p > 0.05$).

**Table 2.** Germination rate in phytotoxicity test.

| | | | | | | | | | | | | |
|---|---|---|---|---|---|---|---|---|---|---|---|---|
| | **Germination Rate (%)** | | | | | | | | | | | |
| | Raw leachate (dilutions in percent) | | | | | | | | | | | |
| Seed | | | Neutral pH | | | | | | Unadjusted pH (8.4) | | | |
| | Control | 2.5 | 10 | 30 | 60% | 90% | Control | 2.5% | 10 | 30 | 60 | 90 |
| *L. sativa* | 97 ± 4 | 97% ± 4 | 97 ± 3 | 52 ± 2 | 24 ± 1 | 2 ± 0.1 | 92 ± 4 | 94 ± 3 | 96 ± 4 | 40 ± 2 | 12 ± 0.5 | 0.8 ± 0.01 |
| *C. sativus* | 97% ± 4 | 97% ± 3 | 97 ± 4 | 62% ± 3 | 23 ± 1 | 2 ± 0.1 | 93 ± 3 | 93 ± 2 | 95 ± 4 | 54 ± 2 | 15 ± 0.5 | 0.0 |
| *P. vulgaris* | 98% ± 4 | 97% ± 4 | 97 ± 4 | 58% ± 3 | 33 ± 1 | 0 | 93 ± 4 | 95 ± 2 | 95 ± 4 | 48 ± 2 | 27 ± 0.5 | 0.0 |
| | Electro-coagulation effluent (dilutions in percent) | | | | | | | | | | | |
| Seed | | | Neutral pH | | | | | | Unadjusted pH (9.5) | | | |
| | Control | 2.5 | 10 | 30 | 60 | 90 | Control | 2.5 | 10 | 30 | 60 | 90 |
| *L. sativa* | 97 ± 4 | 97 ± 4 | 97 ± 4 | 55 ± 0.1 | 26 ± 0.1 | 7 ± 0.3 | 95 ± 4 | 96 ± 3 | 94 ± 4 | 51 ± 2 | 20 ± 1 | 5 ± 0.2 |
| *C. sativus* | 98 ± 4 | 98 ± 4 | 97 ± 4 | 67 ± 0.3 | 28 ± 0.1 | 8 ± 0.4 | 95 ± 3 | 95 ± 3 | 93 ± 4 | 60 ± 3 | 20 ± 1 | 5 ± 0.2 |
| *P. vulgaris* | 98 ± 3 | 97 ± 4 | 98 ± 3 | 63 ± 0.3 | 42 ± 0.2 | 8 ± 0.4 | 95 ± 3 | 93 ± 3 | 95 ± 3 | 60 ± 2 | 38 ± 1 | 5 ± 0.1 |
| | Electro-oxidation effluent (dilutions in percent) | | | | | | | | | | | |
| Seed | | | Neutral pH | | | | | | Unadjusted pH (5.6) | | | |
| | Control | 2.5 | 10 | 30 | 60. | 90. | Control | 2.5 | 10 | 30 | 60 | 90 |
| *L. sativa* | 99 ± 5 | 99 ± 4 | 97 ± 4 | 61 ± 2 | 32 ± 1 | 12.5 ± 0.5 | 98 ± 5 | 98 ± 4 | 97 ± 4 | 57 ± 1 | 29 ± 1 | 10 ± 0.5 |
| *C. sativus* | 100 ± 4 | 100 ± 4 | 98 ± 4 | 72 ± 3 | 40 ± 2 | 15.0 ± 0.5 | 98 ± 4 | 98 ± 3 | 95 ± 4 | 67 ± 3 | 33 ± 1 | 11 ± 0.5 |
| *P. vulgaris* | 100 ± 4 | 98 ± 4 | 100 ± 4 | 72 ± 3 | 53 ± 2 | 21.7 ± 1 | 98 ± 4 | 95 ± 4 | 97 ± 5 | 65 ± 3 | 47 ± 2 | 18 ± 1 |

Results are expressed as mean ± standard deviation.

The germination rate decreased as the leachate content increased, with the exception of the 2.5% and 10% dilutions. At these two concentrations, the germination rates did not show a statistically significant difference, compared with the control tests ($p > 0.05$). From 30% to 90% dilutions, there was a statistically significant decrease in germination rate values in all cases ($p < 0.05$). At 90% dilution of raw leachates with unadjusted pH in the seeds of *P. vulgaris* and *C. sativus*, germination rates of 0% were obtained, and with pH adjustment to neutral, only the seeds of *P. vulgaris* had a germination rate of 0%. It is worth noting that in the case of the treated leachates, there were no germination rates with a value of zero, even at the highest leachate content of 90%. This suggests that the treatment system proposed for leachates was successful in reducing the phytotoxicity.

pH adjustment was found to be an important variable for the germination rate. At a dilution of 30%, the pH adjusted for neutrality generated higher germination rates when compared with the corresponding values of pH without adjustment, obtaining statistically significant values ($p < 0.05$). When comparing the germination rates of the various liquids evaluated, it was discovered that there were statistically significant differences ($p < 0.05$) starting from the 30% dilution, and the order was as follows: raw leachate < EC effluent < EO effluent, for all three seeds and at both pH levels evaluated. This suggests that the decrease in organic matter achieved during the treatment process, in conjunction with any structural changes that may have occurred, played a role in the observed differences in germination rates. The correlation between the enhanced biodegradability index of the treated effluents in the proposed system and the reduction in phytotoxicity appears to be significant.

3.2.2. Effects of Radicle Length and Acquired Biomass on Growth Inhibition

Figures 5 and 6 depict the outcomes obtained for the inhibition of growth based on the length of the radicle and biomass acquired. In the two lowest dilutions (2.5% and 10%), the growth inhibition values were negative in nearly all experiments, suggesting that plant growth was beneficial to some extent in these two dilutions. When comparing the growth inhibition between the dilutions of 2.5 and 10%, there were statistically significant variations in the three seeds evaluated ($p < 0.05$); the dilution of 10% demonstrated improved growth. This was evident, as the most negative growth inhibition values were observed at that particular dilution.

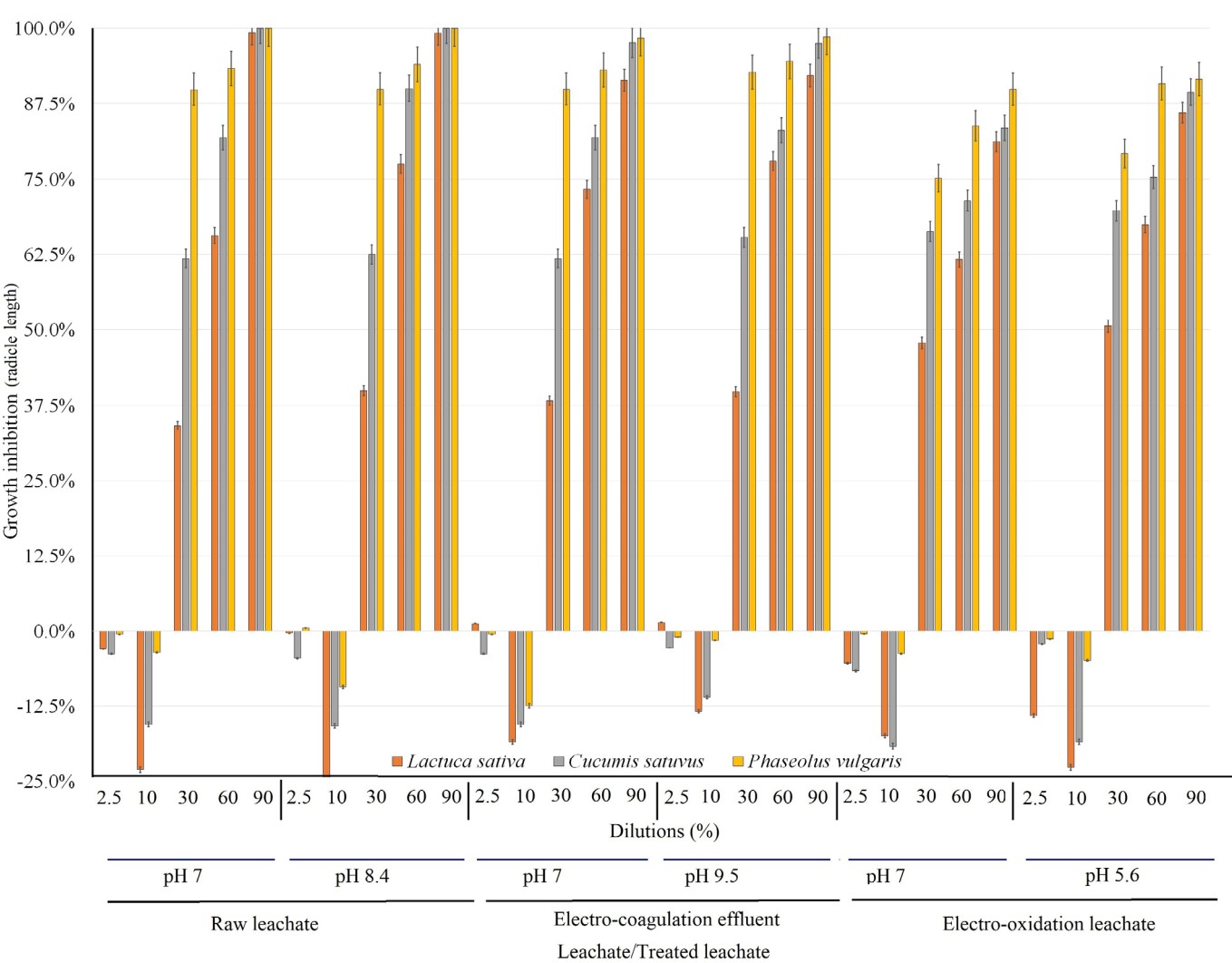

**Figure 5.** Growth inhibition through radicle length in toxicity tests.

When comparing the outcomes of growth inhibition across the three types of leachates, statistically significant differences were observed ($p < 0.05$). The order of these differences in values was inverse to that of the germination rate: crude leachate > EC effluent > EO effluent. Furthermore, the growth inhibition value was negatively correlated with germination; the higher the germination rate, the lower the growth inhibition. These findings demonstrate that the treatment system successfully reduced the phytotoxicity of leachate effluents.

The impact of pH was investigated, and it was discovered that there were statistically significant differences in the growth inhibition values among the three seeds and the three leachates assessed ($p < 0.05$). The unadjusted pH value yielded the highest growth inhibition values, indicating that neutral pH promotes root growth. It is worth mentioning

that citric acid was used to adjust pH, which has been demonstrated to have positive effects on plant growth. Yang and Zhang [90] documented an increase in seedling height, fresh mass, and dry root mass of *P. vulgaris* in response to citric acid applications.

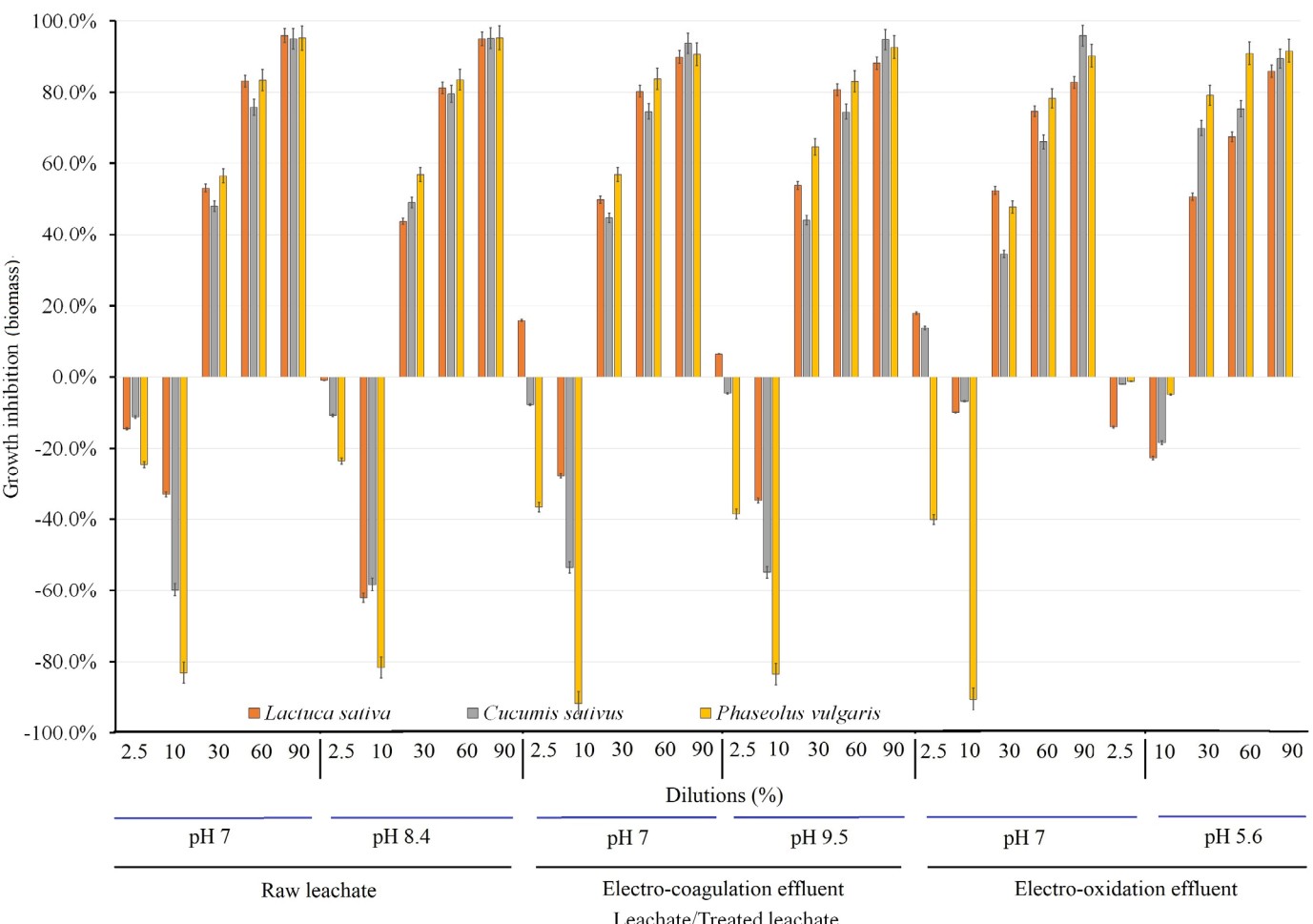

**Figure 6.** Growth inhibition through biomass gained in toxicity tests.

### 3.2.3. Results of Germination Index

The information presented in Figure 7 depicts the germination index values obtained from a series of experiments. The germination rate serves as a measure of the relative germination of seeds, which is determined by the growth of the radicle. This rate indicates the effects of numerous factors on the promotion or inhibition of germination. Notably, the lowest dilutions in all cases, specifically 2.5% and 10%, were statistically different ($p < 0.05$) from the remaining concentrated dilutions. This observation can be attributed to the positive influence of these dilutions on the radicle growth, as previously discussed.

The findings revealed that an increase in leachate content negatively affected the germination rate across all instances, with statistical significance ($p < 0.05$). Conversely, the treatment system had a positive influence on the germination rate of all three seeds evaluated, with statistical significance ($p < 0.05$). Notably, the EO effluent exhibited the highest germination index value, which was statistically significant ($p < 0.05$).

The lowest values for the germination index were recorded for the three seeds in the dilution with the highest concentration of raw leachates (90%) at both pH levels of neutral and pH 8.4. This suggested the presence of phytotoxicity in the leachates, as previously noted by several researchers [91–93]

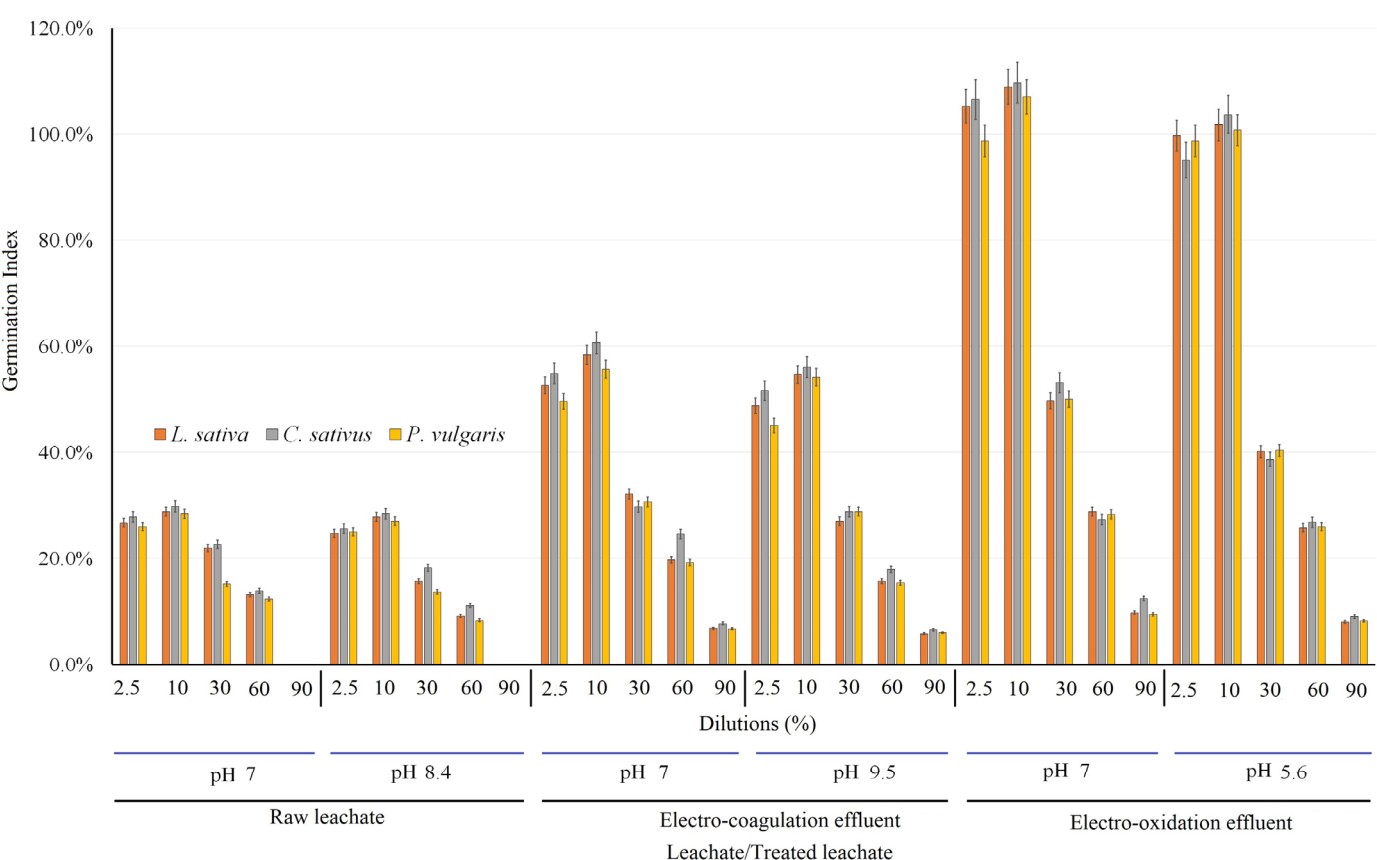

**Figure 7.** Germination index in toxicity tests.

One aspect that must be considered when considering the low germination index values observed in raw leachates is the presence of $NH_3$-N. According to Fuentes [94], elevated concentrations of $NH_3$-N can have detrimental effects on seed germination in various plant species. At the highest concentration of raw leachates, a zero-germination rate may be observed because non-biodegradable organic matter interferes with enzyme activities, which in turn would significantly impact plant growth, as noted by Arunbabu et al. [95].

For the germination index, no values were recorded, even when the leachate concentration reached 90%. As the concentration of leachate increased, a decrease in the germination rate was observed at a dilution of 30%, indicating an increase in phytotoxicity. The pH values of the samples were also analyzed in relation to germination rates, and it was observed that the pH adjusted to neutrality was higher than that without adjustment in all cases ($p < 0.05$). The EO effluent had the highest germination rates at concentrations of 2.5% and 10% of the treated leachates, which were higher than 100%. Obtaining germination rates greater than 100% in phytotoxicity tests with treated leachates has been reported in previous research using Fenton treatment in mature leachates: Poblete et al. [96] using *L. sativa*, and Li et al. [97] with *Z mays*. seeds.

### 3.2.4. Half Maximal Effective Concentration

The half maximal effective concentration is a statistical calculation that represents the expected concentration of a substance in a medium that produces a specific effect on 50% of a given population of organisms under defined conditions [98]. The average effective concentration values obtained from the germination rates in the tests are presented in Table 3.

Upon comparing the mean effective concentration values obtained for various leachates evaluated under identical conditions, it was evident that statistically significant differences existed ($p < 0.05$) in the following descending order: raw leachate < EC effluent < EO effluent.

**Table 3.** Results of median effective concentrations in evaluated seeds.

| | Half Maximal Effective Concentration (%) | |
|---|---|---|
| Seed | Raw leachate | |
| | pH 7 | pH 8.4 |
| *L. sativa* | $2.1 \pm 0.1$ | $1.7 \pm 0.1$ |
| *C. sativus* | $2.3 \pm 0.1$ | $2.0. \pm 0.1$ |
| *P. vulgaris* | $2.9 \pm 0.1$ | $1.6 \pm 0.1$ |
| Seed | Electro-coagulation effluent | |
| | pH 7 | pH 9.5 |
| *L. sativa* | $18.9 \pm 1$ | $16.6 \pm 1$ |
| *C. sativa* | $20.5 \pm 1$ | $18.1 \pm 2$ |
| *P. vulgaris* | $22.8 \pm 2$ | $21.1 \pm 1$ |
| Seed | Electro-oxidation effluent | |
| | pH 7 | pH 5.6 |
| *L. sativa* | $49.8 \pm 2$ | $47.0 \pm 2$ |
| *C. sativus* | $55.7 \pm 2$ | $50.3 \pm 2$ |
| *P. vulgaris* | $62.0 \pm 3$ | $56.2 \pm 2$ |

Results are expressed as mean $\pm$ standard deviation.

An increase in the mean effective concentration is typically accompanied by a decrease in toxicity, as demonstrated by Ward et al. [99]. Notably, studies on leachate treatment systems utilizing the Fenton process have revealed an increase in mean effective concentration values [96,100]. These findings suggest that the proposed treatment system effectively reduces the toxicity of leachates.

The lowest mean effective concentration was noted in the raw leachate with no adjustment of pH in *L. sativa* at $2.1 \pm 0.1$%. Conversely, the highest mean effective concentration value was observed in the EO effluent with a neutral pH in *P. vulgaris* at $62.0 \pm 2.8$%.

The enhancement of the mean effective concentration serves as evidence that the treatment system proposed in this project effectively diminishes the toxicity of the generated effluents. This outcome can be attributed to a substantial reduction of 83% in organic matter, as measured by COD, through the treatment system. Moreover, the biodegradability index demonstrated a significant improvement, increasing from 0.094 in the crude leachate to 0.46 in the EO effluent. This increase in biodegradable content, caused by the oxidation of non-biodegradable material in the EO, positively affects the average effective concentration.

Upon evaluating the average effective concentrations per seed under identical conditions, it was determined that *P. vulgaris* exhibited the highest values, followed by *C. sativa* and *L. sativa*, the latter of which had the lowest values ($p < 0.05$). In the EO effluent, significant differences were observed in the values obtained per seed, with the highest values recorded as 49.82%, 55.72%, and 62.03% for *L. sativa*, *C. sativus*, and *P. vulgaris*, respectively ($p < 0.05$). Additionally, *P. vulgaris* seeds were the least affected by leachates at the mean effective concentration ($p < 0.05$).

pH was found to be a crucial variable in all experiments. Under the same seed and dilution conditions, the mean effective concentration was consistently higher at a neutral pH than at an unadjusted pH ($p < 0.05$). The raw leachate obtained using *P. vulgaris* had a neutral pH of $2.9 \pm 0.1$%, whereas the unadjusted pH resulted in a concentration of $1.6 \pm 0.1$%. When applying the EO effluent with *P. vulgaris*, the values obtained at neutral and unadjusted pH levels were $62.0 \pm 3$ and $56.2 \pm 2$, respectively. A neutral pH promotes germination and growth upon dilution [101]. Additionally, the phytotoxicity results of this study should consider the high salinity levels present in the leachates from the Bordo Poniente landfill. Kalčíková et al. [102] identified high salinity as a contributing factor to leachate toxicity.

### 3.2.5. Transfer Coefficient and Enrichment Coefficient

The transfer and enrichment coefficient data are presented in Tables 4 and 5, respectively. It is generally observed that an increase in the concentration of the effluent generated through EO leads to a corresponding increase in the transfer coefficient. This was attributed to the greater availability of metals resulting from the higher concentration of leachates. Among the three seeds analyzed, calcium had the highest transfer coefficient values. Notably, transfer coefficients lower than one were observed, suggesting that in these cases, the plant did not accumulate the respective metal. Notably, the seeds of *P. vulgaris* exhibited this characteristic, as it was the only seed among the three evaluated that did not accumulate aluminum, even at the highest dilution of leachate.

**Table 4.** Transfer coefficients in experiments conducted.

| Seed | Dilution (%) | Transfer Coefficients | | | | | | | | | |
|------|--------------|------|------|------|------|------|------|------|------|------|------|
| | | **Al** | **Ba** | **Ca** | **Cu** | **Fe** | **K** | **Mg** | **Mn** | **Ni** | **Zn** |
| *L. sativa* | 0 | 0.81 | 0.44 | 1.33 | 0.65 | 0.13 | 0.85 | 0.35 | 0.24 | 0.11 | 0.78 |
| | 2.5 | 0.87 | 0.46 | 1.36 | 0.65 | 0.14 | 0.90 | 0.37 | 0.25 | 0.11 | 0.82 |
| | 10 | 0.93 | 0.48 | 1.36 | 0.70 | 0.15 | 0.95 | 0.39 | 0.26 | 0.11 | 0.87 |
| | 30 | 0.95 | 0.52 | 1.34 | 0.87 | 0.15 | 1.00 | 0.35 | 0.21 | 0.12 | 0.93 |
| | 60 | 0.99 | 0.54 | 1.41 | 0.91 | 0.15 | 1.03 | 0.36 | 0.22 | 0.12 | 0.96 |
| | 90 | 1.00 | 0.54 | 1.43 | 0.91 | 0.16 | 1.04 | 0.36 | 0.23 | 0.14 | 0.97 |
| *C. sativus* | 0 | 0.76 | 0.62 | 3.12 | 0.78 | 0.06 | 0.58 | 0.39 | 0.18 | 0.05 | 0.67 |
| | 2.5 | 0.91 | 0.63 | 3.09 | 0.87 | 0.07 | 0.66 | 0.40 | 0.19 | 0.05 | 0.68 |
| | 10 | 1.08 | 0.78 | 3.13 | 0.87 | 0.10 | 0.72 | 0.42 | 0.15 | 0.05 | 0.69 |
| | 30 | 1.13 | 0.95 | 3.10 | 0.91 | 0.11 | 0.87 | 0.34 | 0.16 | 0.07 | 0.71 |
| | 60 | 1.25 | 1.04 | 3.14 | 0.91 | 0.12 | 0.95 | 0.35 | 0.15 | 0.08 | 0.75 |
| | 90 | 1.37 | 1.15 | 3.15 | 0.91 | 0.13 | 1.05 | 0.37 | 0.22 | 0.09 | 0.82 |
| *P. vulgaris* | 0 | 0.20 | 0.45 | 0.91 | 0.39 | 0.02 | 0.41 | 0.17 | 0.06 | 0.05 | 0.56 |
| | 2.5 | 0.21 | 0.51 | 0.93 | 0.39 | 0.02 | 0.44 | 0.17 | 0.06 | 0.05 | 0.65 |
| | 10 | 0.23 | 0.57 | 1.00 | 0.61 | 0.02 | 0.53 | 0.18 | 0.07 | 0.05 | 0.70 |
| | 30 | 0.27 | 0.57 | 0.98 | 0.61 | 0.03 | 0.52 | 0.14 | 0.09 | 0.06 | 0.71 |
| | 60 | 0.30 | 0.59 | 1.04 | 0.70 | 0.03 | 0.58 | 0.16 | 0.10 | 0.06 | 0.77 |
| | 90 | 0.33 | 0.66 | 1.06 | 0.74 | 0.03 | 0.63 | 0.18 | 0.11 | 0.07 | 0.86 |

Results are expressed as mean ± standard deviation.

**Table 5.** Enrichment coefficient in experiments conducted.

| Seed | Dilution (%) | Enrichment Coefficient | | | | | | | | | |
|------|--------------|------|------|------|------|------|------|------|------|------|------|
| | | **Al** | **Ba** | **Ca** | **Cu** | **Fe** | **K** | **Mg** | **Mn** | **Ni** | **Zn** |
| *L. sativa* | 2.5 | 1.07 | 1.05 | 1.03 | 1.00 | 1.05 | 1.05 | 1.05 | 1.05 | 1.00 | 1.05 |
| | 10 | 1.15 | 1.09 | 1.02 | 1.07 | 1.11 | 1.11 | 1.11 | 1.10 | 1.07 | 1.11 |
| | 30 | 1.16 | 1.19 | 1.01 | 1.33 | 1.12 | 1.17 | 0.98 | 0.90 | 1.13 | 1.18 |
| | 60 | 1.22 | 1.23 | 1.06 | 1.40 | 1.16 | 1.21 | 1.03 | 0.95 | 1.13 | 1.22 |
| | 90 | 1.23 | 1.23 | 1.08 | 1.40 | 1.16 | 1.21 | 1.03 | 0.97 | 1.27 | 1.24 |
| *C. sativus* | 2.5 | 1.20 | 1.02 | 0.99 | 1.11 | 1.09 | 1.13 | 1.05 | 1.01 | 1.00 | 1.02 |
| | 10 | 1.41 | 1.25 | 1.01 | 1.11 | 1.66 | 1.23 | 1.08 | 0.80 | 1.00 | 1.03 |
| | 30 | 1.49 | 1.52 | 1.00 | 1.17 | 1.80 | 1.48 | 0.88 | 0.87 | 1.43 | 1.06 |
| | 60 | 1.64 | 1.67 | 1.01 | 1.17 | 1.98 | 1.63 | 0.92 | 0.84 | 1.57 | 1.12 |
| | 90 | 1.80 | 1.85 | 1.01 | 1.17 | 2.18 | 1.80 | 0.96 | 1.18 | 1.71 | 1.23 |
| *P. vulgaris* | 2.5 | 1.04 | 1.14 | 1.03 | 1.00 | 1.32 | 1.07 | 1.00 | 1.07 | 1.00 | 1.17 |
| | 10 | 1.13 | 1.27 | 1.10 | 1.56 | 1.37 | 1.29 | 1.09 | 1.24 | 1.00 | 1.26 |
| | 30 | 1.33 | 1.27 | 1.08 | 1.56 | 1.59 | 1.28 | 0.83 | 1.59 | 1.29 | 1.28 |
| | 60 | 1.46 | 1.32 | 1.15 | 1.78 | 1.75 | 1.41 | 0.97 | 1.72 | 1.29 | 1.39 |
| | 90 | 1.61 | 1.48 | 1.17 | 1.89 | 1.92 | 1.55 | 1.12 | 1.90 | 1.43 | 1.54 |

Results are expressed as mean ± standard deviation.

The enrichment coefficient values for all cases revealed that the concentration of metals in contaminated soils was higher than that in the baseline scenario, implying greater accumulation of these metals in plants grown in such soils than in the baseline scenario. Values greater than one were obtained, indicating a higher concentration of metals in contaminated soils. As the concentration of leachates increased, the enrichment coefficient also increased. The maximum value attained was 1.92 for iron, with a leachate concentration of 90%.

### 3.3. Analysis Using Inductively Coupled Plasma-Optical Emission Spectroscopy (ICP-OES)

Metals and metalloids in the soil are prevalent in both urban and rural settings and pose a persistent threat to the environment. Unlike organic molecules, these substances do not break down and, therefore, require careful attention. Given that many sanitary landfills are situated near crop areas, it is imperative to conduct phytotoxicity tests on the raw and treated leachates. In the Bordo Poniente landfill (Mexico City) Stage IV employs a 1 mm HDPE geomembrane as a waterproofing solution [103,104]

#### 3.3.1. Samples of Used Garden Soil

Table 6 presents the findings of the *ICP-OES* analysis of the garden land employed in this study, along with a parallel examination of the recommended values put forth by the New York State Department of Environmental Conservation (NYSDEC) [105].

**Table 6.** Results of *ICP-OES* analysis of garden soil used in phytotoxicity tests and comparison with recommended values.

| Parameter | PQL | Wavelength (nm) | Concentration (mg kg$^{-1}$) | Maximum Recommended Value (mg kg$^{-1}$) * | Parameter | PQL | Wavelength (nm) | Concentration (mg kg$^{-1}$) | Maximum Recommended Value (mg kg$^{-1}$) * |
|---|---|---|---|---|---|---|---|---|---|
| Ag | 0.6 | 328 | <0.6 | | K | 62.5 | 766 | 35,600 ± 10 | |
| Al | 6.3 | 396 | 3201 ± 110 | | Mg | 0.6 | 280 | 10,615 ± 550 | |
| As | 6.3 | 194 | <6 | 16 | Mn | 0.6 | 258 | 486 ± 10 | |
| Ba | 0.6 | 233 | 98 ± 4 | 350 | Ni | 3.1 | 232 | 140 | |
| Be | 0.6 | 313 | <0.6 | | Pb | 6.3 | 220 | 7 ± 0.20 | 400 |
| Ca | 15.6 | 318 | 2268 ± 110 | | Se | 6.3 | 196 | <6 | |
| Cd | 0.6 | 226 | 2 ± 0.05 | 2.5 | Ti | 0.6 | 336 | 1184 ± 50 | |
| Co | 3.1 | 229 | 13 ± 0.06 | | Tl | 6.3 | 191 | <6 | |
| Cu | 3.1 | 327 | 23 ± 1 | 270 | V | 0.6 | 312 | 50 ± 2 | |
| Fe | 3.1 | 238 | 24,125 ± 1120 | | Zn | 0.6 | 2104 | 97 ± 4 | 2200 |

PQL: Practical Quantitation Limit. * Recommended maximum values for agricultural soil as recommended by NYSDEC [105].

According to the suggested maximum limits for metal content in cultivation soil stipulated by the NYSDEC [105], the levels of metals present in the garden soil utilized in this research project were below the aforementioned maximum limits. This observation validated the suitability of the soil used in the phytotoxicity assays and confirmed that it did not have any adverse impact on the test outcomes.

#### 3.3.2. Samples of Roots

The findings obtained through *ICP-OES* analysis of the roots during phytotoxicity testing proved to be highly enlightening, shedding light on the interplay between the elements absorbed by the roots and their potential effects on growth inhibition and germination rates. As shown in Table 7, root characterization was facilitated by *ICP-OES*. Elements below the practical limit of quantification were discarded.

**Table 7.** *ICP-OES analysis of roots collected in phytotoxicity tests.*

| | Parameter | Al | Ba | Ca | Cu | Fe | K | Mg | Mn | Ni | Zn |
|---|---|---|---|---|---|---|---|---|---|---|---|
| Seed | Wavelength (nm) | 393 | 233 | 318 | 327 | 238 | 766 | 280 | 258 | 232 | 214 |
| | PQL | 6.3 | 0.6 | 15.6 | 3.1 | 3.1 | 62.5 | 0.6 | 0.6 | 3.1 | 0.6 |
| | Dilution (%) | Concentration (mg kg$^{-1}$) | | | | | | | | | |
| *L. sativa* | 0 | 2599 ± 130 | 43 ± 2 | 3013 ± 150 | 15 ± 1 | 3222 ± 160 | 30,399 ± 1500 | 3739 ± 140 | 115 ± 5 | 15 ± 1 | 76 ± 3 |
| | 2.5 | 2788 ± 150 | 45 ± 2 | 3093 ± 140 | 15 ± 1 | 3391 ± 140 | 31,999 ± 1400 | 3936 ± 160 | 121 ± 6 | 15 ± 1 | 80 ± 2 |
| | 10 | 2987 ± 180 | 47 ± 2 | 3082 ± 160 | 16 ± 1 | 3570 ± 120 | 33,683 ± 1600 | 4143 ± 210 | 127 ± 5 | 16 ± 1 | 84 ± 3 |
| | 30 | 3025 ± 150 | 51 ± 1 | 3037 ± 140 | 20 ± 1 | 3609 ± 120 | 35,452 ± 110 | 3681 ± 140 | 104 ± 4 | 17 ± 1 | 90 ± 3 |
| | 60 | 3176 ± 140 | 53 ± 1 | 3189 ± 120 | 21 ± 1 | 3739 ± 110 | 36,724 ± 120 | 3865 ± 150 | 109 ± 5 | 17 ± 1 | 93 ± 4 |
| | 90 | 3192 ± 150 | 53 ± 1 | 3247 ± 150 | 21 ± 1 | 3753 ± 160 | 36,851 ± 110 | 3848 ± 160 | 111 ± 5 | 19 ± 2 | 94 ± 4 |
| *C. sativus* | 0 | 2442 ± 120 | 61 ± 3 | 7067 ± 380 | 18 ± 1 | 1457 ± 1.1 | 20,805 ± 80 | 4094 ± 210 | 89 ± 4 | 7 ± 1 | 65 ± 3 |
| | 2.5 | 2919 ± 80 | 62 ± 3 | 7018 ± 270 | 20 ± 1 | 1591 ± 0.6 | 23,552 ± 110 | 4287 ± 180 | 90 ± 4 | 7 ± 1 | 66 ± 3 |
| | 10 | 3442 ± 150 | 76 ± 6 | 7108 ± 480 | 20 ± 1 | 2417 ± 110 | 25,532 ± 120 | 4413 ± 120 | 71 ± 3 | 7 ± 1 | 67 ± 2 |
| | 30 | 3631 ± 140 | 93 ± 4 | 7035 ± 300 | 21 ± 1 | 2626 ± 90 | 30,868 ± 150 | 3595 ± 140 | 77 ± 4 | 10 ± 4 | 69 ± 2 |
| | 60 | 3994 ± 120 | 102 ± 4 | 7113 ± 320 | 21 ± 1 | 2889 ± 160 | 33,955 ± 140 | 3754 ± 170 | 75 ± 4 | 11 ± 5 | 73 ± 3 |
| | 90 | 4393 ± 200 | 113 ± 5 | 7139 ± 410 | 21 ± 1 | 3178 ± 180 | 37,350 ± 160 | 3950 ± 200 | 105 ± 5 | 12 ± 5 | 80 ± 3 |
| *P. vulgaris* | 0 | 654 ± 40 | 44 ± 2 | 2058 ± 140 | 9 ± 1 | 422 ± 300 | 14,557 ± 700 | 1753 ± 120 | 29 ± 1 | 7 ± 3 | 54 ± 2 |
| | 2.5 | 682 ± 30 | 50 ± 2 | 2117 ± 160 | 9 ± 1 | 557 ± 40 | 15,518 ± 100 | 1753 ± 180 | 31 ± 1 | 7 ± 3 | 63 ± 2 |
| | 10 | 736 ± 30 | 56 ± 2 | 2264 ± 100 | 14 ± 1 | 578 ± 30 | 18,838 ± 110 | 1910 ± 80 | 36 ± 1 | 7 ± 2 | 68 ± 2 |
| | 30 | 868 ± 40 | 56 ± 2 | 2225 ± 110 | 14 ± 1 | 670 ± 20 | 18,627 ± 80 | 1447 ± 110 | 46 ± 2 | 9 ± 2 | 69 ± 3 |
| | 60 | 955 ± 40 | 58 ± 2 | 2358 ± 80 | 16 ± 1 | 737 ± 30 | 20,489 ± 1000 | 1692 ± 120 | 50 ± 2 | 9 ± 4 | 75 ± 3 |
| | 90 | 1050 ± 50 | 65 ± 2 | 2413 ± 120 | 17 ± 1 | 811 ± 40 | 22,538 ± 1000 | 1961 ± 140 | 55 ± 2 | 10 ± 1 | 83 ± 4 |

PQL: Practical Quantitation Limit. Values expressed as mean ± standard deviation.

Metals and metalloids, when present in high concentrations, are persistent toxins in organisms because of their inability to be degraded and irreversible immobilization in the environment. Excessive accumulation of these substances can impair physiological processes, such as photosynthesis and the synthesis of chlorophyll pigments. Upon exposure to heavy metals, plants generate reactive oxygen species (ROS) as a primary response [106]. ROS are naturally produced as byproducts of plant cell metabolism, but their excessive production due to environmental stressors leads to oxidative damage and eventual cell death [107]. Heavy metals can induce ROS production directly through the Haber-Weiss reaction, which generates hydroxyl radicals (•OH) from $H_2O_2$ (hydrogen peroxide) and superoxide ($O_2^-$), or indirectly through their own toxicity [108]. ROS accumulation occurs when there is an imbalance between ROS production and the activity of the antioxidant system. These highly toxic molecules can oxidize biological macromolecules such as lipids, proteins, and nucleic acids, leading to lipid peroxidation, membrane damage, and enzyme inactivation [106]. The specific contributions of the evaluated elements to toxicity are briefly described.

- Although aluminum (Al) is not considered an essential nutrient, it has agronomic significance because of its toxic effects on plants [109]. Al toxicity was first observed in the root systems of plants, which are particularly vulnerable to the toxic effects of Al. Al inhibits root elongation and cell division, resulting in poor growth and reduced plant development [110].
- Barium (Ba) can be considered to have a slightly detrimental effect on plant growth as it competes with calcium, which is necessary for plant growth. However, this effect would only occur if the Ba levels in the soil exceeded the recommended maximum [105].
- Copper (Cu) is an essential micronutrient for plants in small amounts. However, excessive Cu in the growing medium can have a detrimental impact on root development by burning its tips, leading to excessive lateral growth, reduced branching, and ultimately plant decline [111]. Excessive Cu can also cause chlorosis, a condition that negatively affects plant growth and development [112].
- Iron (Fe) plays a fundamental role in plant growth and is involved in various biochemical processes, including respiration, chlorophyll synthesis, pathogen defense, the generation and elimination of reactive oxygen species, and photosynthesis. Both Fe deficiency and excess result in harmful effects on plant development such as chlorosis [112].

- Nickel (Ni), another essential micronutrient for plant growth, plays a crucial role in enzymatic catalysis as a component of various compounds. However, excessive Ni can negatively affect plant growth by affecting enzyme function [113].
- Zinc (Zn) is frequently present in insoluble forms in the soil and serves as an essential micronutrient for plants. Nevertheless, excessive levels of Zn can have deleterious effects on plant growth, as revealed by recent research [114]. It plays a critical role in the synthesis of carbohydrates during photosynthesis and in the metabolism of hormones by regulating the levels of auxins (a plant hormone that promotes plant growth and development).
- Calcium (Ca) is a structural element in plants that is present in the cell wall and membrane and plays a fundamental role in cell division and elongation [115]. Ca deficiency symptoms are commonly observed in growing organs, including apical meristems, which promote growth when plants germinate [116].
- Magnesium (Mg) plays a critical role in plant metabolism and its mobility within plants is highly beneficial for growth. As a fundamental component of chlorophyll, Mg enables plants to effectively perform photosynthesis, making it essential for crop health and productivity [117].
- Potassium (K), a crucial plant nutrient, plays an indispensable role in plant health and development. As a significant macronutrient, it comprises the majority of inorganic cations in plants and accounts for 10% of plant dry weight. This essential nutrient is primarily sourced from the soil [118].
- Manganese (Mn) is an essential micronutrient crucial for the proper functioning of various plant processes, including root cell elongation. Plants can actively absorb Mn in the form of $Mn^{2+}$, but excessive levels of Mn can have detrimental effects on plant growth and development by replacing Mg in enzymatic reactions [119].
- Excessive amounts of potassium and manganese can lead to toxicity, resulting in the inhibition of the absorption of other essential nutrients, such as Ca, Fe, and Zn, as reported in recent studies [117,118].

The primary objective of this research was not to assess the phytostabilization capacity of the analyzed seeds, as no timeframes were set to enable the complete development of the plant. Nonetheless, the phytostabilization effect attained during the initial stages of seed growth merits consideration. Phytostabilization involves the application of plants with a high tolerance to metals to polluted soils, mechanically stabilizing them and thereby limiting the movement of pollutants through the action of plant roots [120]. Phytostabilization involves multiple mechanisms, the first of which involves furnishing litter and vegetation cover to decrease leaching by increasing the soil's water storage capacity and promoting evapotranspiration. Moreover, trees aid in controlling erosion and creating an aerobic environment in the rhizosphere, while also adding organic matter that enhances soil aggregation and binds to contaminants. In addition, heavy metals with low mobility, such as Pb, can accumulate in significant amounts in root tissues and effectively function as a form of phytostabilization. Phytostabilization can also be attained through metal immobilization outside the roots due to root exudates, which causes the metals to precipitate [120].

A direct relationship was observed between the evaluated metals (Al, Ba, Cu, Fe, Ni, and Zn) and the leachate content for all metals. Specifically, the higher the concentration of leachates, the higher the metal content in all cases, which was statistically significant ($p < 0.05$) for all dilutions. It was also observed that *P. vulgaris* has a lower retention capacity for the evaluated metals, as it obtained the lowest values for Al, Cu, Fe, Ni, and Zn. In contrast, *C. sativus* had the highest retention capacities for Al, Ba, Cu, and Zn.

Chen et al. [121] suggested that *L. sativa* adapts to oxidative stress caused by Al through polyphenolic biosynthesis, which could have been observed in the phytotoxicity tests of this project, particularly in the case of EO effluent, where the lowest values of growth inhibition were obtained, compared to crude leachate and EC effluent. The highest Al content in the roots was observed in *C. sativus*, at $4393 \pm 200$ mg kg$^{-1}$ with 90% raw

leachate, and the lowest was $654 \pm 40$ mg kg$^{-1}$ with *P. vulgaris* in the control sample. Al may have been a negative contributor to plant development during the phytotoxicity tests.

The highest level of Ba recorded in this study was in the roots of *C. sativus*, which had a raw leachate content of 90% and a level of $113 \pm 8$ mg kg$^{-1}$. This level did not exceed the recommended maximum value of 350 mg kg$^{-1}$ [105], and it can be concluded that Ba did not have a negative impact on the phytotoxicity tests conducted.

In the current study, the highest Cu content was recorded at $21 \pm 3$ mg kg$^{-1}$ in *C. sativus* with 90% crude leachate, which exceeded the recommended maximum of 10 mg kg$^{-1}$ [111]. This finding implies that Cu is another element impeding the growth of the analyzed seeds. It is worth noting that the Cu content in the seeds, with the exception of the roots of *C. sativus* with 2.5% raw leachate and the control sample at 9 mg kg$^{-1}$, was above the recommended maximum limit.

In these tests, the maximum Fe value was observed in *C. sativus* at 90% of the raw leachate at $3753 \pm 160$ mg kg$^{-1}$, which was much higher than the recommended maximum of 200 mg kg$^{-1}$ [112]. The Fe content in all evaluated roots was above the maximum recommended value. Although it is true that phytotoxicity tests carried out in this project did not involve the evaluation of chronic effects on plant growth, in the case of chlorosis, it can be considered that this excess iron provided by the leachates negatively influenced the growth of the evaluated seeds.

The highest concentration of Ni was observed in *L. sativa*, at $19 \pm 5$ mg kg$^{-1}$, and the recommended maximum value for this element in the roots was 22 mg kg$^{-1}$ [113]. Interestingly, no negative effects on phytotoxicity were observed when the Ni content was below the recommended maximum for all the evaluated roots.

In the case of Zn, suitable values for plant development ranged from 15 to 20 mg kg$^{-1}$ Zn [114]. The highest concentration of Zn in the analyses was presented in *L. sativa* at $94 \pm 4$ mg kg$^{-1}$ with 90% raw leachate, four times the appropriate value. Therefore, Zn could be considered to have contributed negatively to the phytotoxicity test results, as it was higher than those recommended for all the evaluated roots.

No direct correlation was observed between leachate dilution and root content for Ca, K, Mg, and Mn. For instance, the roots of *C. sativus* in the control sample had a Ca concentration of $7067 \pm 380$ mg kg$^{-1}$, whereas in the sample with 90% crude leachate, the concentration was $7139 \pm 410$ mg kg$^{-1}$, with both values not statistically significant ($p > 0.05$).

It was not possible to identify the recommended maximum values for Ca in plants in the literature, as it is a structural element of the plants themselves. Therefore, there were no quantifiable elements to determine whether Ca played a role in the phytotoxicity of the evaluated leachates.

According to the *ICP-OES* analysis of the roots, potassium levels exceeded the rest, with *L. sativa* at 90% leachate registering $36 \pm 1$ g kg$^{-1}$ and the *P. vulgaris* control sample showing $14 \pm 0.7$ g kg$^{-1}$. Upon examining the potassium content of the seeds, it was observed that *P. vulgaris* roots had the lowest potassium content ($p < 0.05$). Given that plant potassium sufficiency levels range from 10 to 50 g of potassium per kg of dry matter [122], it can be inferred that potassium does not adversely affect leachate phytotoxicity.

Analysis of the Mg values of the evaluated seeds revealed that the roots of *P. vulgaris* had the lowest values ($p < 0.05$). According to the literature, Mg sufficiency values in plants range from 2 to 10 g per kg of dry mass [123,124]. As the Mg values were below the maximum range, it can be inferred that Mg did not adversely affect the phytotoxicity of the leachates.

When comparing the Mg values by seed, it was observed that the root of *P. vulgaris* had the lowest value ($p < 0.05$). According to the literature, Mg sufficiency values in plants range from 50 to 150 mg per kg of dry mass [125,126]. All Mn values in the roots were below the maximum value, indicating that Mn did not influence leachate toxicity.

## 4. Conclusions

- The characterization of the leachates employed in this research project indicates that they possess a biodegradability index of 0.094, a chemical oxygen demand of $3.4 \pm 0.1 \text{ g L}^{-1}$, a dissolved organic carbon of $1.2 \pm 0.05 \text{ gL}^{-1}$, a color of $3200 \pm 100$ Pt-Co U, and a $NH_3$-N content of $0.66 \pm 0.03 \text{ gL}^{-1}$. Consequently, it can be asserted that the leachates in question are mature.
- The parameters for the enhanced elimination of organic matter, as measured by COD, were established for both EC and EO processes. For EC, the optimal current density was found to be $23.3 \text{ mA cm}^{-2}$, with a stirring rate of 120 revolutions per minute and a pH of 7. For EO, the conditions were determined to be a NaCl concentration of $1.0 \text{ g L}^{-1}$, an electrode distance of 0.75 cm, a current density of $33.3 \text{ mA cm}^{-2}$, and a pH of 7.
- Under conditions of greater removal of organic matter, measured as COD, removal values were reached in the chemical demand of oxygen, dissolved organic carbon, color, and $NH_3$-N in the EC process of 63%, 69%, 94%, and 50%, respectively. For the EO process, these values were 82, 86, 99, and 81%, respectively.
- The proposed treatment system resulted in a significant enhancement of biodegradable organic matter. The concentration of biodegradable COD increased from 26% in the raw leachate to 39% following the EC process and further increased to 58% in the effluent of the EO process. Additionally, the biodegradability index, which was initially 0.094 in the crude leachate, improved to 0.26 with the EC process and attained a value of 0.46, following the EO process.
- The concentration of particulate COD in the EC effluent decreased from 48% to 23%. The EC process effectively removed colloidal species that could have impeded the subsequent EO process, demonstrating its suitability as an initial treatment stage.
- The conversion of a portion of the recalcitrant organic matter present in raw leachates into biodegradable materials and $CO_2$ was achieved through both EC and EO processes. These processes resulted in a significant alteration in the chemical structure of the recalcitrant organic matter.
- By analyzing the organic matter content in an EC and EO system used to treat mature leachates, the structural changes that enhance the biodegradability of the resulting wastewater were uncovered.
- Based on the data collected in this study, it can be concluded that the parameters that significantly contributed to the toxicity in the leachates examined were aluminum, copper, iron, and zinc.
- The findings of the phytotoxicity assessments indicated that the proposed treatment approach led to a diminution of the phytotoxicity of the effluents produced. This outcome can be ascribed to alterations in the molecular composition of the organic matter.

## 5. Recommendations

- In this investigation, it was found that garden soil was a consistent factor in all of the experiments conducted. Therefore, future phytotoxicity trials should assess the effects of different soil types, such as sandy and clayey soil, to better understand the relationship between plants and the chemical composition of leachates in the soil.
- A valuable area of inquiry is to evaluate the influence of emerging pollutants on phytotoxicity tests. Although these pollutants have received considerable attention in recent times, the cessation of operations at the Bordo landfill in 2012 restricts the applicability of this variable. Therefore, it is proposed that phytotoxicity tests incorporate leachates from landfills with nearer closure dates, or even those that continue to function.

**Author Contributions:** A.M.-C.: investigation; conceptualization; methodology; performed the experiments; analyzed and interpreted the data; writing—original draft preparation. M.N.R.-V.: validation; formal analysis; resources; contributed reagents; conceived and designed the experiments; analysis tools or data; writing—review and editing; supervision; project administration; funding acquisition. All authors have read and agreed to the published version of the manuscript.

**Funding:** This research was supported by the UNAM project. Project No. R-275, Name of the project "Investigation Lines under Dra. María Neftalí Rojas-Valencia".

**Data Availability Statement:** Data is contained within the article.

**Acknowledgments:** The authors thank the USI-IIUNAM for help in finding some references.

**Conflicts of Interest:** The authors declare no conflicts of interest. The funders had no role in the design of the study; in the collection, analyses, or interpretation of data; in the writing of the manuscript; or in the decision to publish the results.

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
