# Peer review of "Assessment of Phytotoxicity in Untreated and Electrochemically Treated Leachates through the Analysis of Early Seed Growth and Inductively Coupled Plasma-Optical Emission Spectroscopy Characterization"

_horticulturae, doi:10.3390/horticulturae10010067_

Round 1
Reviewer 1 Report
Comments and Suggestions for Authors
-
General comments:
-
The study presents an effective treatment approach for stabilized leachates, employing electro-coagulation (EC) and electro-oxidation (EO) technologies. The results indicate substantial removal rates for various parameters, with a notable increase in biodegradability and reduction in phytotoxicity. The research addresses an important environmental concern and contributes valuable insights to wastewater treatment.
-
Specific comments:
-
- 1) The improvement in biodegradability, particularly the increase from 26% to 58% in the electro-oxidation (EO) process effluent, is a noteworthy finding. Further details on the mechanisms responsible for this enhancement would enhance the comprehension of the study.
- 2) The removal rates for COD, color, and nitrogen are impressive. Providing information on the specific conditions or parameters optimized during electro-coagulation (EC) and electro-oxidation (EO) would be beneficial for potential applications.
- 3) The reduction in humic and fulvic acids during both EC and EO processes is highlighted. Exploring the implications of these changes on the overall environmental impact and water quality could add depth to the discussion.
-
Constructive feedback:
-
- Elaborate on the operational parameters (e.g., current density, treatment duration) for both electro-coagulation and electro-oxidation processes. This information is crucial for replicability and practical implementation.
- Discuss potential challenges or limitations encountered during the application of EC and EO technologies. Addressing these aspects would contribute to a more comprehensive understanding of the treatment system's feasibility and robustness.
- Consider expanding the discussion on the specific roles of aluminum, copper, iron, zinc, and calcium in contributing to toxicity. Insights into their removal mechanisms and implications for environmental safety would enhance the overall impact of the study.
-
Summary:
-
The study successfully demonstrates the efficacy of electro-coagulation and electro-oxidation technologies in treating stabilized leachates. Substantial removal rates for COD, color, and nitrogen, coupled with enhanced biodegradability and reduced phytotoxicity, underscore the practical significance of the proposed treatment system. However, providing more details on operational parameters and addressing potential limitations would further strengthen the research's applicability and impact.
Author Response
Dear reviewer, please see my responses in the attachment.

Reviewer 2 Report
Comments and Suggestions for Authors
Dear Authors,
The paper under consideration addresses a highly relevant and contemporary theme of global significance. Evaluating and analyzing leachates are pivotal macro-themes within the broader context of circular economy practices worldwide. In acknowledging the critical role that leachates play in the environmental impact of various products consumed and used globally, please highlight a topic at the forefront of sustainable waste management and environmental preservation efforts by the relevant schema for the whole process, for example.
However, to fully leverage the potential impact of their research, it is imperative to consider a comprehensive restructuring of the article. The chosen topic holds immense importance, and its depth warrants a more thorough exploration. The introductory section, in particular, should develop into the intricacies of leachate evaluation, emphasizing its relevance in the circular economy context.
Moreover, to enhance the clarity and impact of your findings, please consider reorganizing the sections on electrocoagulation and electrooxidation processes. By clearly delineating the optimal parameters for each treatment method, you can effectively communicate the practical applications of their research, adding the main chemical reactions that occur in these types of processes and, if possible, their underlying mechanism.
The overall paper sounds good, but a few aspects need improvements for readers' credibility.
Below are a few suggestions for the authors' consideration:
- To add a map of the considered study area;
- To add a suitable diagram for the methodology used in the experimental part;
- Please add the minimum information required for the performance parameters of ICP-OES measurements (e.g., the limit of quantitation and the limit of detection for each considered element).
- Please, add the other minimum relevant information required for the metal accumulation in the root of the considered plants (e.g., the transfer coefficient of elements from soil to the root of plants and the enrichment factor, respectively). Also, I recommend arranging the metallic elements into categories, e.g., macroelements, microelements, and trace elements (if applicable).
Other minor considerations:
- For clarity, please specify the meaning of the NH3-N combination (sometimes referring to ammonium ion and other times to molecular nitrogen).
- Be aware that there are two figures numbered with the digit 1.
- R88 – add at least one name of a low-molecular-weight acid to which you refer.
- R180 – please reformulate for clarity: The dilutions were prepared by mixing raw leachates and treated leachates with tap water under conditions that maximized COD removal by EC and EO. COD is a calculated indicator that reflects the content of organic matter in an analyzed water sample and cannot be eliminated in a physical/chemical process. (At least, that's how I understood it!)
- R 201-204 – please remove the dots between the calculation formulas and the equation numbers, respectively, and correctly format the equations mathematically.
- R654-656 – How was this conclusion reached, considering that the degree of aromaticity of the evaluated organic chemical compounds is not discussed in the context of the article?
Also, I recommend following the journal's author guidelines, reviewing the citation style in the text, and reformatting the list of bibliographic references according to the requirements.
Author Response

(The authors gave the same response as above.)

Round 2
Reviewer 1 Report
Comments and Suggestions for Authors
The text emphasizes the efficacy of electro-coagulation (EC) and electro-oxidation (EO) in addressing stabilized leachates. These methods notably diminished COD, enhanced biodegradability from 26% to 58%, reduced particulate COD, and mitigated phytotoxicity. As a reviewer, I find this acceptable.
Author Response
We appreciate the time devoted by Reviewer 1 to the publication of this scientific article.
Reviewer 2 Report
Comments and Suggestions for Authors
Dear Authors,
The article revision has strengthened the quality and clarity of your work, making it a much more substantial contribution to the field, and it is evident that you have put a significant amount of effort into addressing the reviewers' comments.
While the manuscript is now in good shape, a few technical editing aspects could benefit further attention. Please pay more attention to the numbering of the figures and tables (e.g., there are two figures marked as 1 and two as 3). Additionally, I have noticed that lines 623-627 are without text.
With a final round of careful editing and attention to these minor details, your article will be ready for publication. It will make a valuable contribution to the field.
Warm regards,
Author Response
Dear Reviewer, please see the answers in the attachment.
